# Structural basis of nucleosomal histone H4 lysine 20 methylation by SET8 methyltransferase

Cheng-Han Ho[1,2,*], Yoshimasa Takizawa[1,*], Wataru Kobayashi[1,3], Yasuhiro Arimura[1], Hiroshi Kimura[4] , Hitoshi Kurumizaka[1,2,3]

SET8 is solely responsible for histone H4 lysine-20 (H4K20) mono-methylation, which preferentially occurs in nucleosomal H4. However, the underlying mechanism by which SET8 specifically promotes the H4K20 monomethylation in the nucleosome has not been elucidated. Here, we report the cryo-EM structures of the human SET8–nucleosome complexes with histone H3 and the centromeric H3 variant, CENP-A. Surprisingly, we found that the overall cryo-EM structures of the SET8–nucleosome complexes are substantially different from the previous crystal structure models. In the complexes with H3 and CENP-A nucleosomes, SET8 specifically binds the nucleosomal acidic patch via an arginine anchor, composed of the Arg188 and Arg192 residues. Mutational analyses revealed that the interaction between the SET8 arginine anchor and the nucleosomal acidic patch plays an essential role in the H4K20 monomethylation activity. These results provide the groundwork for understanding the mechanism by which SET8 specifically accomplishes the H4K20 monomethylation in the nucleosome.

## Introduction

Chromatin accommodates genomic DNA in eukaryotes. The fundamental unit of chromatin is the nucleosome, which wraps DNA around a histone octamer, containing two copies of each of the core histones H2A, H2B, H3, and H4 (Wolffe, 1998). Posttranslational modifications (PTMs) of histones, such as methylation, acylation, and phosphorylation, predominantly occur in the N-terminal tails of histones, and function as epigenetic marks to recruit their specific binding proteins to chromatin (Strahl & Allis, 2000; Kouzarides, 2007; Bannister & Kouzarides, 2011). These modification-specific histone-binding proteins, termed "readers," regulate genomic DNA accessibility by changing the higher order structure and dynamics of chromatin (Ruthenburg et al, 2007). In contrast, histone-modifying enzymes, termed "writers," demarcate genomic regions with distinct chromatin structures and functions by introducing specific histone PTMs, which recruit their particular binding proteins (Ruthenburg et al, 2007).

Among histone PTMs, methylation of histone H4 lysine-20 (H4K20) plays pivotal roles in the DNA damage response, DNA replication, mitotic chromosome condensation, and transcription regulation, and is therefore crucial for genome maintenance (Brustel et al, 2011; Wu & Rice, 2011; Beck et al, 2012; Jørgensen et al, 2013). The H4K20 residue is mono-, di-, and trimethylated, and the di- and trimethylations are only promoted on monomethylated H4K20, and not on the unmethylated H4K20 residue (Brustel et al, 2011; Beck et al, 2012; Jørgensen et al, 2013). H4K20 dimethylation is introduced on the previously monomethylated H4K20 residue by the Suv4-20 family of methyltransferases (Southall et al, 2014).

SET8 (also named PR-SET7, SETD8, or KMT5A) is a histone methyltransferase that is solely responsible for H4K20 monomethylation in cells (Oda et al, 2009; Brustel et al, 2011; Beck et al, 2012; Jørgensen et al, 2013). Intriguingly, SET8 primarily promotes H4K20 monomethylation in the nucleosome (Fang et al, 2002; Nishioka et al, 2002), although it also possesses the ability to methylate the nucleosome-free H4K20 residue (Couture et al, 2005). The nucleosome containing the centromeric histone H3 variant, CENP-A, is a preferred substrate for SET8 because the H4 N-terminal tails in the CENP-A nucleosome are more accessible than those in the canonical H3 nucleosome (Arimura et al, 2019). However, the mechanism by which SET8 specifically targets the nucleosome has remained elusive.

In the present study, we determined the structures of the human SET8–nucleosome complexes with histone H3 and CENP-A by cryo-EM at 3.15 and 3.00 Å resolutions, respectively. The structures explain how SET8 specifically methylates the H4K20 residue in nucleosomes.

## Results

### The cryo-EM structures of the SET8–nucleosome complexes containing histone H3 and CENP-A

To clarify the mechanism by which SET8 specifically promotes H4K20 monomethylation in the nucleosome, we performed single-particle

[1]Laboratory of Chromatin Structure and Function, Institute for Quantitative Biosciences, The University of Tokyo, Tokyo, Japan    [2]Department of Biological Sciences, Graduate School of Science, The University of Tokyo, Tokyo, Japan    [3]Graduate School of Advanced Science and Engineering, Waseda University, Tokyo, Japan    [4]Cell Biology Center, Institute of Innovative Research, Tokyo Institute of Technology, Yokohama, Japan

Correspondence: kurumizaka@iqb.u-tokyo.ac.jp
*Cheng-Han Ho and Yoshimasa Takizawa contributed equally to this work

cryo-EM. We purified full-length human SET8 as a recombinant protein (Figs 1A and S1A). The nucleosome core particle (NCP) was reconstituted with recombinant human histones, H2A, H2B, H3.1, and H4, in the presence of the 145-base pair Widom 601 DNA (Fig S1B and C) (Lowary & Widom, 1998). The SET8–NCP complex was then purified by sedimentation in the presence of paraformaldehyde (GraFix) (Fig S1D) (Kastner et al, 2008), and visualized by cryo-EM (Fig 1B and C). It is possible that the paraformaldehyde cross-linking may affect the SET8 interaction with the NCP. We processed the SET8–NCP complex, followed by a single-particle workflow in the RELION software package (Kimanius et al, 2016). The cryo-EM structure of the SET8–NCP complex was then determined at 3.15 Å resolution (Figs 1D and S2A–C). Surprisingly, the overall structure of the SET8–NCP complex is different from the structures previously predicted from the low resolution X-ray diffraction data (Fig S3) (Girish et al, 2016). Unexpectedly, the SET8 interaction with

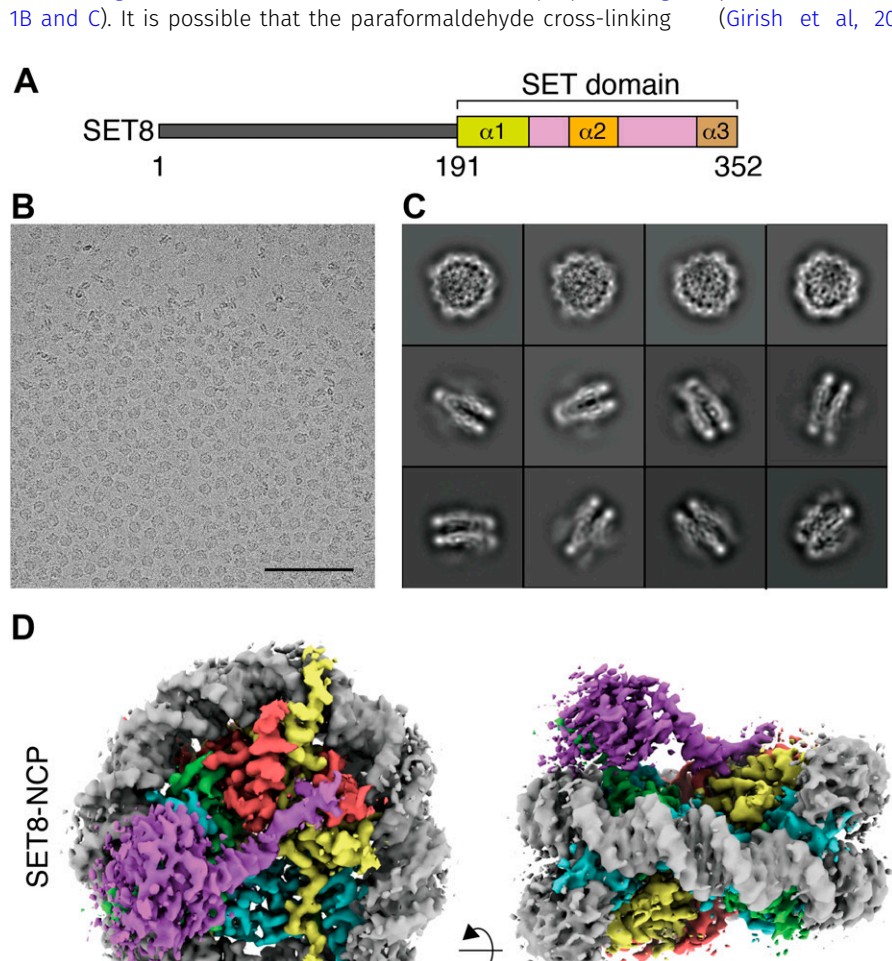

**Figure 1. Cryo-EM structures of the SET8–NCP and SET8–NCP$^{CENP-A}$ complexes.**
**(A)** Schematic representation of the SET8 domain structures. **(B)** Digital micrograph of the SET8–NCP complex. Scale bar, 100 nm. **(C)** Representative 2D class averages of the SET8–NCP complex. Box size, 18.9 nm$^2$. **(D, E)** Cryo-EM reconstructions of the SET8–NCP complex (D) and the SET8–NCP$^{CENP-A}$ complex (E). The cryo-EM maps were visualized by UCSF ChimeraX.

nucleosomal DNA was not obvious, although the EM map of the SET8 region near the nucleosomal DNA is ambiguous.

SET8 can efficiently monomethylate the H4K20 residue in the NCP containing CENP-A (NCP^CENP-A) (Arimura et al, 2019). This may happen because, in the NCP^CENP-A, the N-terminal tails of H4 adopt the outward conformation, which is preferred for the H4K20 monomethylation by SET8 (Arimura et al, 2019). We prepared the SET8–NCP^CENP-A complex (Fig S1A and E–G), and the cryo-EM structure was determined at 3.00 Å resolution (Figs 1E and S2D–F). Consistent with the previous structural studies with CENP-A NCPs (Tachiwana et al, 2011; Pentakota et al, 2017; Chittori et al, 2018; Tian et al, 2018; Allu et al, 2019; Yan et al, 2019; Zhou et al, 2019), the DNA regions of the nucleosomal entry/exit sites are somewhat ambiguous because of the flexible nature of the DNA in the NCP^CENP-A (Fig S4A and B). The structure of the CENP-A–specific RG loop is also different from the corresponding loop of H3 in the NCP (Fig S4C and D). However, the overall structure of the SET8–NCP^CENP-A complex is quite similar to that of the SET8–NCP complex. Therefore, the structural characteristics of the NCP^CENP-A may not affect the specific NCP binding by SET8, except for its preferred H4 N-terminal tail conformation.

### The SET8 N-terminal arginine anchor binds the acidic patch of the nucleosome

In both the SET8–NCP and SET8–NCP^CENP-A complexes, the globular SET domain of SET8 is located on the surface of the histone octamer

(Fig 1D and E). The crystal structure of the SET domain (Couture et al, 2005; Girish et al, 2016) fits well into the EM density maps. Interestingly, the α1 helix of the SET domain is clearly observed in the SET8–NCP and SET8–NCP^CENP-A complexes (Fig 2). We found that the N-terminal extension of the SET8 α1 helix forms the arginine anchor, in which the Arg188 and Arg192 residues bind to the acidic patch of the NCP (Fig 2A and B). The SET8 Arg188 and Arg192 residues are separately captured by two acidic pockets, formed with the H2A Glu56 and H2B Glu113 residues and the H2A Glu61 and Glu92 residues, respectively (Fig 2B). The EM densities of the Arg188 and Arg192 residues are also clearly visible in the SET8 N-terminal extension bound to the NCP^CENP-A, and both residues are captured by the acidic patch (Fig 2C and D). Therefore, SET8 recognizes both canonical and CENP-A NCPs by the same mechanism, using the Arg188 and Arg192 residues in the N-terminal arginine anchor.

### The SET8 arginine anchor is important for its H4K20 monomethylation activity in the nucleosome

The acidic patch of the NCP functions as the binding platform for many NCP binding proteins (McGinty & Tan, 2016). To test the functional importance of acidic patch binding by SET8, we prepared an acidic patch-defective NCP (NCP^apd). In the NCP^apd, the acidic patch Glu and Asp residues (H2A Glu56, H2A Glu61, H2A Glu64, H2A

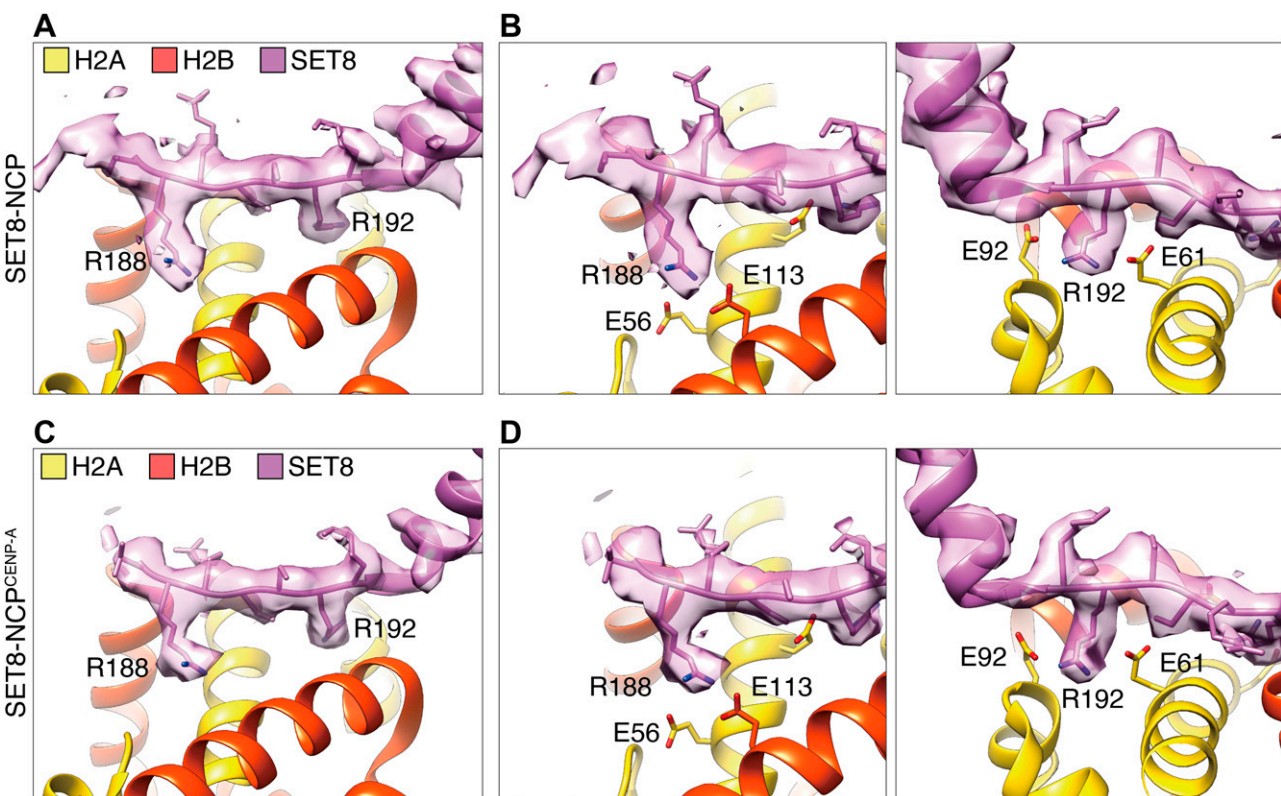

**Figure 2. Arginine anchor of SET8 bound to the acidic patches of the nucleosome core particle (NCP) and the NCP^CENP-A.**
**(A, B, C, D)** Close-up views of the SET8 Arg188 and Arg192 residues bound to the H2A Glu56 and H2B Glu113 residues (B, D, left) and the H2A Glu61 and Glu92 residues (B, D, right), respectively, in the acidic patches of the NCP (A, B) and the NCP^CENP-A (C, D). The atomic model of SET8 (PDB: 1ZKK) was docked into the EM density maps of the SET8–NCP and SET8–NCP^CENP-A complexes.

Asp90, H2A Glu91, H2A Glu92, H2B Glu105, and H2B Glu113) were replaced by neutral, hydrophilic Thr, and Ser residues, respectively (Kujirai et al, 2020). These amino acid replacements in H2A and H2B did not affect NCP formation (Fig S5A and B). Interestingly, SET8–NCP binding was drastically decreased in the NCP[apd], as compared with the wild-type NCP (Fig 3A, lanes 1–6, Figs 3B and S6). Consistently, H4K20 monomethylation was undetectable in the NCP[apd] (Fig 3C, lanes 1–6, and Fig S7). We also prepared a SET8 mutant, SET8 R188A/R192A, in which both arginine anchor residues, Arg188 and Arg192, were replaced by Ala (Fig S5C). As expected, the NCP binding of the SET8 R188A/R192A mutant was substantially reduced (Fig 3A, lanes 7–12, Figs 3B and S6), and the H4K20 monomethylation was also diminished (Fig 3C, lanes 7–9, and Fig S7). These results indicated that interaction between the acidic patch and the SET8 arginine anchor plays an essential role in the H4K20 monomethylation activity on the NCP.

## The peptide-binding cleft of SET8 captures the H4 N-terminal tail in the nucleosome

In the SET8–NCP complexes, the cryo-EM densities corresponding to the H4 N-terminal tails of the NCP and the NCP[CENP-A] were clearly observed (Fig 4A and B). The superimposition of the crystal structure of the SET domain on the cryo-EM maps of the SET8–NCP complex or the SET8–NCP[CENP-A] complex revealed that the H4 N-terminal tail fits very well within the peptide-binding cleft of SET8 (Fig 4A and B). In the canonical NCP, the H4 N-terminal tail reportedly adopts two configurations, outward and inward (Arimura et al, 2019). In the NCP[CENP-A], the outward conformation of the H4 N-terminal tail is preferred (Arimura et al, 2019). Intriguingly, in both the SET8–NCP and SET8–NCP[CENP-A] complexes, the H4 N-terminal tail adopts the outward configuration and is incorporated within the peptide-binding cleft of the SET domain (Fig 4C). Therefore, we

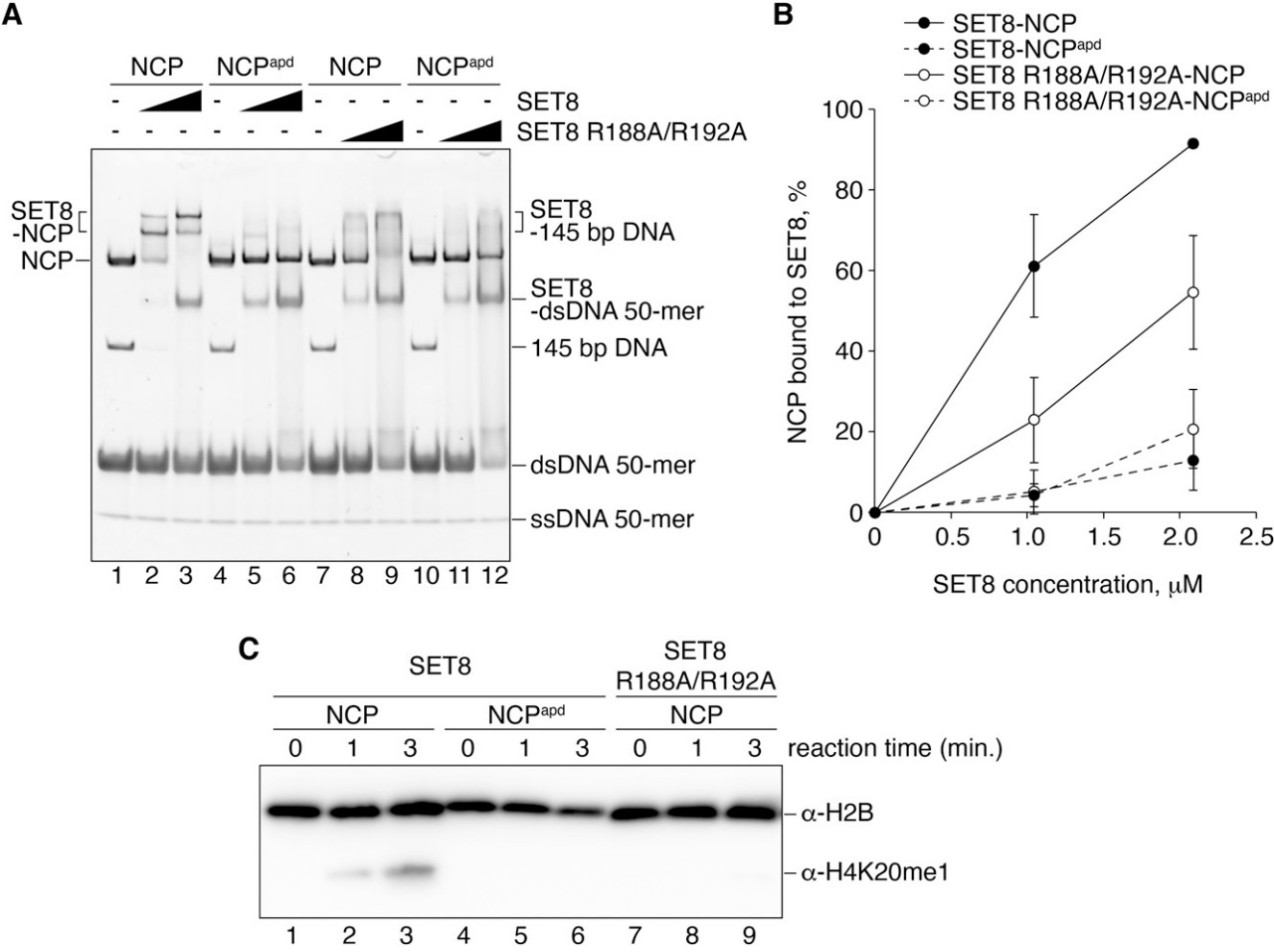

**Figure 3. Mutational analyses for the SET8–NCP complex interaction.**
**(A)** Gel shift assay of the NCP or the NCP[apd] (the acidic patch-defective nucleosome) with SET8 and SET8 R188A/R192A. Double-stranded DNA and single-stranded DNA are denoted as dsDNA and ssDNA, respectively. The amount of SET8 was titrated. A double-stranded DNA 50-mer containing a trace amount of single-stranded 50-mer was included as competitor DNA. NCP (0.52 $\mu$M; lanes 1–3, 7–9) and NCP[apd] (0.52 $\mu$M; lanes 4–6, 10–12) were mixed with SET8 (0, 1.0, and 2.1 $\mu$M; lanes 1 and 4, 2 and 5, and 3 and 6, respectively) or SET8 R188A/R192A (0, 1.0, and 2.1 $\mu$M; lanes 7 and 10, 8 and 11, and 9 and 12, respectively). **(A, B)** Quantification of the results in (A). The average % values of three independent experiments shown in Figs 3A and S6 are plotted against the SET8 concentration. **(C)** Time course methylation assay of the NCP with SET8 or SET8 R188A/R192A, and the NCP[apd] with SET8. Lanes 1–3, 4–6, and 7–9 indicate results for the NCP with SET8, the NCP[apd] with SET8, and the NCP with SET8 R188A/R192A, respectively. The experiments were repeated three times, and the reproducibility was confirmed (Fig S7).

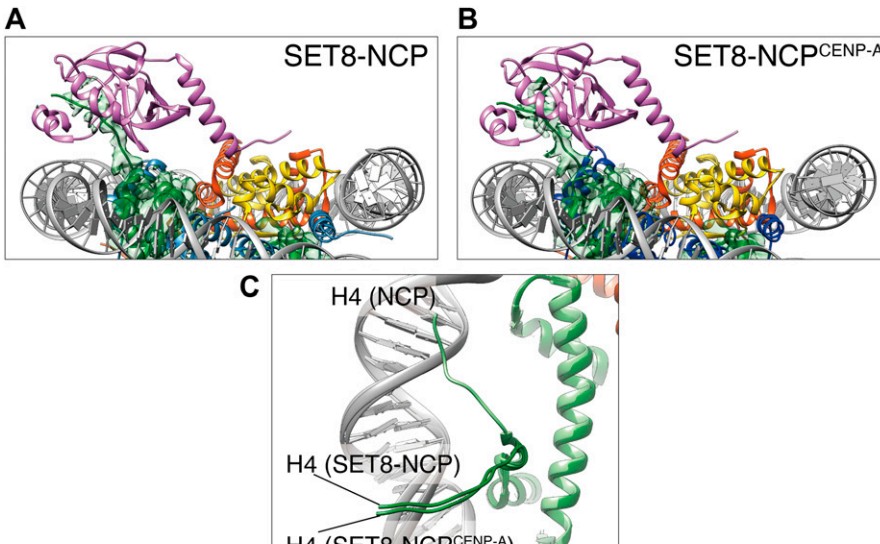

**Figure 4. Interaction of SET8 with the H4 N-terminal tails of the nucleosome core particle (NCP) and the NCP^CENP-A.**
**(A, B)** Close-up views of the H4 N-terminal tail bound to the SET domain. **(C)** Comparison of the H4 N-terminal tail conformations in the SET8–NCP complex, the SET8–NCP^CENP-A complex, and the NCP (PDB: 5Y0C). Histone H3, CENP-A, and histone H4 are colored light blue, blue, and light green, respectively. Three structures are superimposed.

concluded that the outward configuration of the H4 N-terminal tail is actually the preferred substrate for SET8.

## Discussion

H4K20 monomethylation primarily occurs on the nucleosome, rather than the nucleosome-free H4 (Nishioka et al, 2002; Fang et al, 2002). To understand the mechanism by which the H4K20 residue is specifically monomethylated in the nucleosome, in the present study, we determined the cryo-EM structures of the SET8–NCP and SET8–NCP^CENP-A complexes (Fig 1). We found that SET8 contains an arginine anchor formed by Arg188 and Arg192, which specifically binds to the acidic patch of the nucleosomes (Fig 2). Mutational analyses revealed that the interaction between the SET8 arginine anchor and the nucleosomal acidic patch is pivotal in SET8–NCP binding and H4K20 monomethylation (Fig 3). These findings explain why SET8 specifically monomethylates the nucleosomal H4K20 residue, rather than the nucleosome-free H4 (Fang et al, 2002; Nishioka et al, 2002). The acidic patch binding by the SET8 arginine anchor may dictate the SET domain orientation on the nucleosome and, thus, fix the peptide-binding cleft of the SET domain in the appropriate position to accommodate the H4 N-terminal tail of the nucleosome (Fig 4).

The present structure differs substantially from the crystal structure model, in which SET8 binds nucleosomal DNA (Fig S3). The previous crystallographic analysis was performed with the acidic patch binding protein, RCC1, to facilitate crystallization (Girish et al, 2016). The presence of RCC1 may inhibit the proper binding of the SET8 arginine anchor to the nucleosomal acidic patch, and perturb active SET8–NCP complex formation. Meanwhile, the acidic patch blocking by RCC1 may help to capture a transient nucleosomal DNA binding state of SET8. This short-lived SET8–NCP binding may function in properly positioning SET8 on the NCP surface, where the SET8 arginine anchor and catalytic center bind to the acidic patch and the H4 N-terminal tail, respectively (Figs 1, 2, and 4).

Cryo-EM structures of a histone H3 methyltransferase, DOT1L, complexed with a nucleosome-containing ubiquitinated H2B have been reported (Anderson et al, 2019; Jang et al, 2019; Valencia-Sanchez et al, 2019; Worden et al, 2019; Yao et al, 2019). DOT1L is the human homolog of yeast Dot1, which promotes mono-, di-, and trimethylations of the H3 Lys79 residue (H3K79) in the nucleosome (Nguyen & Zhang, 2011), and belongs to a different class of lysine methyltransferases from SET8. In fact, the Dot1-class proteins lack the SET domain, which is the catalytic domain of SET8. However, DOT1L binds the nucleosomal acidic patch by a mechanism similar to that of SET8, using the arginine anchor containing two conserved arginine residues. The H2B ubiquitination at the Lys120 residue reportedly enhances the DOT1L methyltransferase activity (Briggs et al, 2002; Ng et al, 2002; McGinty et al, 2008). The H2BK120 ubiquitination reportedly stabilizes the DOT1L–nucleosome binding, but does not affect the overall location of DOT1L on the nucleosome surface (Yao et al, 2019). This suggests that the H2BK120 ubiquitination may function as an auxiliary factor in the DOT1L-mediated H3K79 methylation. Collectively, the evolutionarily conserved arginine anchor in different classes of histone methyltransferases, SET8 and DOT1L, plays a pivotal role in proper nucleosome binding via the acidic patch, and may ensure accurate lysine methylation of the H4K20 and H3K79 residues, respectively.

Previous crystal structures of the NCPs revealed that the H4 N-terminal tail has two conformations, the inward and outward configurations (Arimura et al, 2019). In the SET8–NCP and SET8–NCP^CENP-A complexes, the H4 N-terminal tails are close to the outward configuration (Fig 4), indicating that this configuration is required for the H4K20 monomethylation by SET8. In contrast, the inward configuration of the H4 N-terminal tail is induced by its binding to the acidic patch of the neighboring NCP (Luger et al, 1997). The SET8 arginine anchor may have an additional function to mask the acidic patch of the nucleosome, and thus shift the nucleosomal H4 N-terminal tail to the outward configuration for efficient H4K20 monomethylation.

# Materials and Methods

## Purification of histones

Histones H2A, H2B, H3.1, H4, H2A E56T/E61T/E64T/D90S/E91T/E92T (H2A$^{apd}$), and H2B E105T/E113T (H2B$^{apd}$) were prepared as described previously (Kujirai et al, 2018). Briefly, N-terminally His$_6$-tagged H2A, H2B, H3.1, H4, H2A$^{apd}$, and H2B$^{apd}$ were expressed in *Escherichia coli* cells. The cells were lysed, and the insoluble fraction was collected and denatured in buffer, containing 50 mM Tris–HCl (pH 8.0), 7 M guanidine-HCl, 500 mM NaCl, and 5% glycerol. The His$_6$-tagged histones were then purified by Ni-NTA agarose chromatography (QIAGEN) under denaturing conditions. Subsequently, the His$_6$-tags were removed by thrombin protease cleavage, and the histones were purified by Mono S column chromatography (GE Healthcare). Finally, the histones were dialyzed against water, freeze-dried, and stored at 4°C.

## Preparation of histone octamers

The H2A-H2B-H3.1-H4 octamer was reconstituted with the freeze-dried histones H2A, H2B, H3.1, and H4 at a 1:1:1:1 stoichiometry, and denatured in 20 mM Tris–HCl buffer (pH 7.5), containing 7 M guanidine-HCl and 2 mM 2-mercaptoethanol. The mixture was then dialyzed against refolding buffer, containing 10 mM Tris–HCl (pH 7.5), 2 M NaCl, 1 mM EDTA, and 5 mM 2-mercaptoethanol. Finally, the histone octamer was purified by HiLoad 16/600 Superdex 200 pg (GE Healthcare) column chromatography and stored at –80°C. The H2A$^{apd}$-H2B$^{apd}$-H3.1-H4 octamer was prepared similarly with the freeze-dried histones H2A$^{apd}$, H2B$^{apd}$, H3.1, and H4. The H2A-H2B-CENP-A-H4 octamer was also prepared similarly with the freeze-dried histones H2A, H2B, CENP-A, and H4, except that it was denatured in 20 mM Tris–HCl buffer (pH 7.5), containing 7 M guanidine-HCl and 20 mM 2-mercaptoethanol.

## Reconstitution of NCP

Canonical NCP, NCP$^{CENP-A}$, and NCP$^{apd}$ were reconstituted by the salt dialysis method, with the 145-base pair Widom 601 DNA and a histone octamer containing H2A-H2B-H3.1-H4, H2A-H2B-CENP-A-H4, or the H2A$^{apd}$-H2B$^{apd}$-H3.1-H4 (Tachiwana et al, 2010). After dialysis, the NCPs were purified with a Prep Cell Model 491 apparatus (Bio-Rad), using a native polyacrylamide gel. The NCPs were eluted in TCS buffer, which contains 20 mM Tris–HCl (pH 7.5) and 1 mM dithiothreitol.

## SET8 and SET8 R188A/R192A purification

Human SET8 (KMT5A Isoform 2; Uniprot ID: Q9NQR1-2) was used in this study. The coding region of SET8 was inserted into a modified pET15b vector, which contains a His$_6$-tag and a PreScission Protease recognition sequence, instead of the thrombin recognition sequence. SET8 was purified according to the previously described method (Arimura et al, 2019). Briefly, human SET8 was produced in *E. coli* BL21(DE3) by induction with isopropyl β-D-1-thiogalactopyranoside. The cells were sonicated, and the supernatant containing His$_6$-tagged SET8

was collected by centrifugation. His$_6$-tagged SET8 was purified by Ni-NTA affinity chromatography. The concentration of the recovered His$_6$-tagged SET8 was measured by the Bradford method. PreScission protease was added to the sample (4 U/mg), which was then dialyzed against Mono S wash buffer, containing 50 mM Tris–HCl (pH 7.5), 100 mM NaCl, 10% glycerol, and 2 mM 2-mercaptoethanol. Precipitates were removed by centrifugation. After confirming the His$_6$-tag removal by SDS–PAGE, SET8 was further purified by Mono S cation exchange chromatography with Mono S elution buffer, containing 50 mM Tris–HCl (pH 7.5), 600 mM NaCl, 10% glycerol, and 2 mM 2-mercaptoethanol. SET8 was finally purified by HiLoad 16/600 Superdex 200 pg (GE Healthcare) gel filtration chromatography, in buffer containing 20 mM Tris–HCl (pH 7.5), 100 mM KCl, 0.2 mM EDTA, 10% glycerol, and 1 mM DTT. The purified SET8 was stored at –80°C. The plasmid for the production of SET8 R188A/R192A was generated by PCR site-directed mutagenesis, and SET8 188A/192A was also purified similarly.

## Gradient fixation (GraFix) of the SET8–NCP and SET8–NCP$^{CENP-A}$ complexes

The SET8–NCP sample was prepared by mixing NCP (0.52 $\mu$M) with SET8 (1.0 $\mu$M) in 1 ml of reaction solution, containing 10 mM HEPES-KOH (pH 7.8), 18 mM Tris–HCl (pH 7.5), 50 mM NaCl, 50 mM KCl, 0.10 mM EDTA, 5% glycerol, 0.5 mM DTT, and 0.10 mM S-adenosyl-L-homocysteine. SET8 was separately added four times to the reaction solution containing the NCP. The sample mixture was incubated at 25°C in a water bath for 5 min after each SET8 addition. After the SET8 addition was completed, the sample mixture was incubated at 25°C in a water bath for 15 min. The SET8–NCP$^{CENP-A}$ sample was also prepared similarly, by mixing NCP$^{CENP-A}$ (0.52 $\mu$M) with SET8 (0.94 $\mu$M) in 1 ml of reaction solution.

The gradient was prepared with a Gradient Master (SKB). Buffer 1 (10 mM HEPES-NaOH [pH 7.5], 100 mM NaCl, 1 mM DTT, and 5% sucrose) and buffer 2 (10 mM HEPES-NaOH [pH 7.5], 100 mM NaCl, 1 mM DTT, 20% sucrose, and 4% paraformaldehyde) were added to Ultra-Clear Centrifuge Tubes (344058; Beckman Coulter), using the short cap. The gradient was prepared by the SHORT Sucrose 5–20% (SW28) method, and then cooled at 4°C for 1 h.

Before applying the samples onto the gradient solution in the centrifuge tube, 1 ml of the gradient solution was removed from the top. After applying the samples onto the gradient solution, the centrifuge tubes were placed in an SW32Ti swinging bucket rotor (Beckman Coulter) and centrifuged at 27,000 rpm at 4°C for 16 h. The sample fractions (1 ml each) were collected from the top of the gradient solution and analyzed by 6% native polyacrylamide gel electrophoresis with 0.2× (TBE) Tris-Borate-EDTA buffer. The fractions containing the SET8–NCP or the SET8–NCP$^{CENP-A}$ complex were then purified by chromatography on PD-10 columns (GE Healthcare), in 10 mM HEPES–NaOH buffer (pH 7.5) containing 2 mM TCEP (pH 7.5). The samples were finally concentrated and stored at 4°C.

## Cryo-EM

For the cryo-EM specimen preparation of both the SET8–NCP and the SET8–NCP$^{CENP-A}$ complexes, the samples (2.5 $\mu$l) were applied to glow-discharged grids (Quantifoil R1.2/1.3 200-mesh Cu). The grids were blotted without wait time for 5 s (SET8–NCP) or 6 s

(SET8–NCP^CENP-A) with the bolt force set to 0, under 100% humidity at 4°C, using a Vitrobot Mark IV (Thermo Fisher Scientific), and were then directly plunged into liquid ethane. Both the SET8–NCP and SET8–NCP^CENP-A complexes were recorded on a Krios G3i cryo-electron microscope (Thermo Fisher Scientific), operated at 300 kV. For the SET8–NCP complex, 2,355 movies were recorded using the EPU (Thermo Fisher Scientific) auto acquisition software with a pixel size of 1.05 Å. Digital micrographs of the SET8–NCP complex were recorded with 63 s exposure times on a Falcon 3EC (Thermo Fisher Scientific) direct electron detector in the electron counting mode, retaining a total of 51 frames with a total dose of ~52 electron/Å². For the SET8–NCP^CENP-A complex, 6,075 movies were recorded using the SerialEM (Mastronarde, 2005) auto acquisition software with a pixel size of 1.05 Å. Digital micrographs of the SET8–NCP^CENP-A complex were recorded with 7 s exposure times on a K3 BioQuantum (Gatan) direct electron detector in the electron counting mode, using a slit width of 25 eV and retaining 40 frames with a total dose of ~60 electron/Å².

## Image processing

All movie frames of both the SET8–NCP and SET8–NCP^CENP-A complexes were aligned using MOTIONCOR2 (Zheng et al, 2017) with dose weighting. The contrast transfer function (CTF) estimation was performed by CTFFIND4 (Rohou & Grigorieff, 2015) from digital micrographs with dose weighting. For the following image processing of both the SET8–NCP and SET8–NCP^CENP-A complexes, RELION3.0 and RELION3.1 (Zivanov et al, 2018) were used. The particles were semi-automatically picked with a box size of 180 × 180 pixels, and junk particles were removed by 2D classification, followed by 3D classification. The crystal structure of the NCP (3LZ0), low-pass–filtered to 60 Å, was used as the initial model for the 3D classification of the SET8–NCP complex. The ab initio model generated in the RELION3.1 was used as the initial model for the 3D classification of the SET8–NCP^CENP-A complex. The 3D classifications for both the SET8–NCP and SET8–NCP^CENP-A complexes were performed, followed by particle polishing and a few rounds of CTF refinement. The resolutions of the refined 3D maps of the SET8–NCP and the SET8–NCP^CENP-A complexes were at 3.15 and 3.00 Å, respectively, as estimated by the gold standard Fourier Shell Correlation (FSC) at an FSC = 0.143 (Scheres, 2016). Local resolutions of the SET8–NCP and SET8–NCP^CENP-A complexes were calculated by RELION3.1. The final maps of the SET8–NCP and SET8–NCP^CENP-A complexes were normalized with MAPMAN (Kleywegt et al, 2004), and visualized with UCSF Chimera (Pettersen et al, 2004) and UCSF ChimeraX (Goddard et al, 2018). The details of the processing statistics for the SET8–NCP and SET8–NCP^CENP-A complexes are listed in Table S1.

## Model building and refinement

The crystal structures of the NCP (PDB: 3LZ0), SET8 (PDB: 1ZKK) without ligands, and the atomic model of the CENP-A from NCP^CENP-A (PDB: 6C0W) were placed in the cryo-EM maps of the SET8–NCP and SET8–NCP^CENP-A complexes, by rigid-body fitting in UCSF Chimera (Pettersen et al, 2004). The complete models of the SET8–NCP and SET8–NCP^CENP-A complexes were manually built with COOT (Emsley &

Cowtan, 2004), followed by real-space refinement in Phenix (Adams et al, 2010).

## Gel shift assay

Purified NCP or NCP^apd (0.52 μM) was mixed with SET8 or SET8 R188A/R192A (1.0 and 2.1 μM) in 5.0 μl of reaction solution, containing 10 mM HEPES-KOH (pH 7.8), 16 mM Tris–HCl (pH 7.5), 50 mM NaCl, 50 mM KCl, 0.12 mM EDTA, 5% glycerol, 0.5 mM DTT, 0.10 mM S-adenosyl-L-homocysteine, and 1.5 μM double-stranded DNA 50-mer (as competitor DNA). For comparison, the double-stranded DNA 50-mer (1.5 μM) or 145-base pair DNA (0.52 μM) was mixed with SET8 (2.1 and 4.2 μM) in 5 μl of reaction solution, containing 10 mM HEPES-KOH (pH 7.8), 16 mM Tris–HCl (pH 7.5), 50 mM NaCl, 50 mM KCl, 0.12 mM EDTA, 5% glycerol, 0.5 mM DTT, and 0.10 mM S-adenosyl-L-homocysteine. The mixed solutions were then incubated at 25°C in a water bath for 30 min. The samples were separated by 6% non-denaturing polyacrylamide gel electrophoresis with 0.2× TBE buffer. The gels were stained with EtBr and imaged with an Amersham Imager 680 (GE Healthcare).

## Nucleosomal H4K20 monomethylation assay

Purified NCP or NCP^apd (0.52 μM) was mixed with SET8 or SET8 R188A/R192A (0.15 μM) in 5.0 μl of reaction solution, containing 10 mM HEPES-KOH (pH 7.8), 10 mM Tris–HCl (pH 7.5), 50 mM NaCl, 20 mM KCl, 60 μM EDTA, 2% glycerol, 0.50 mM DTT, 80 μM S-adenosylmethionine, and 1.5 μM double-stranded DNA 50-mer (as a competitor). The reaction mixtures were incubated for 1 or 3 min at 25°C. The reaction was stopped by adding 5 μl of 4% SDS solution, containing 0.10 mM Tris–HCl (pH 6.8), 20% glycerol, and 0.2% bromophenol blue. The samples were then heated at 95°C for 15 min before fractionation by SDS-18% polyacrylamide gel electrophoresis, using a gel prepared with WIDE RANGE gel preparation buffer (Nacalai Tesque). After the electrophoresis, the proteins were transferred to an Amersham Hybond 0.2 μm (PVDF) polyvinylidene difluoride membrane (GE Healthcare) by a Trans-Blot SD Semi-Dry Transfer Cell (Bio-Rad). The membrane was blocked by 5% skim milk powder dissolved in phosphate buffered saline containing 0.05% Tween 20 (PBS-T) for 1 h at room temperature. The membrane was washed with PBS-T and incubated with the primary antibodies, the mouse monoclonal antibody against monomethylated H4K20 (CMA421; 32 Hayashi-Takanaka et al, 2015) and the anti-H2B monoclonal antibody (53H3: Cell Signaling), diluted with Can Get Signal solution 1 (TOYOBO) at 4°C overnight. The anti-monomethylated H4K20 antibody was diluted to a final concentration of 1 μg/ml, and the anti-H2B antibody was diluted 10,000-fold. The membrane was washed with PBS-T, and then incubated with the secondary antibody (Amersham ECL) and mouse IgG, HRP-linked F(ab')₂ fragment from sheep (NA9310: GE Healthcare) diluted 10,000-fold with Can Get Signal solution 2 (TOYOBO) at 4°C for 2 h. The membrane was washed with PBS-T, and Amersham ECL Prime Western Blotting Detection Reagent (GE Healthcare) was added to the membrane. The image of the blot was acquired by chemiluminescent detection using an Amersham Imager 680 (GE Healthcare).

# Data Availability

The cryo-EM reconstructions and atomic models of the SET8–NCP complex and the SET8–NCP^CENP-A complex have been deposited in the Electron Microscopy Data Bank under the accession codes EMD-30551 and EMD-30552, and the Protein Data Bank under the accession codes PDB: 7D1Z and PDB: 7D2O, respectively.

# Supplementary Information

# Acknowledgements

We thank Y Iikura (University of Tokyo), Y Takeda (University of Tokyo), and H Ishii (Waseda University) for their assistance, and Dr. PA Wade (National Institute of Environmental Health Sciences) for critical reading. We also thank M Kikkawa (University of Tokyo) for cryo-EM data collection. Funding: This work was supported in part by (JSPS) Japan Society for the Promotion of Science KAKENHI Grant Numbers JP17H01408 (to H Kurumizaka), JP18H05534 (to H Kurumizaka), JP18H05527 (to H Kimura), JP19K06522 (to Y Takizawa), (JST) Japan Science and Technology Agency CREST Grant Number JPMJCR16G1 (to H Kurumizaka and H Kimura), the Platform Project for Supporting Drug Discovery and Life Science Research (BINDS) from (AMED) Japan Agency for Medical Research and Development under Grant Number JP20am0101076 (to H Kurumizaka), and JP20am0101115j0004 (to M Kikkawa), and JST ERATO Grant Number JPMJER1901 (to H Kurumizaka).

## Author Contributions

CH Ho: data curation, validation, investigation, visualization, methodology, and writing—original draft, review, and editing.
Y Takizawa: data curation, funding acquisition, validation, investigation, visualization, methodology, and writing—original draft, review, and editing.
W Kobayashi: investigation.
Y Arimura: investigation.
H Kimura: resources and funding acquisition.
H Kurumizaka: conceptualization, supervision, funding acquisition, and writing—original draft, review, and editing.

## Conflict of Interest Statement

The authors declare that they have no conflict of interest.

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
