## [Reviewer comments · Life Science Alliance]

Life Science Alliance

Structural basis of nucleosomal histone H4 lysine 20 methylation by SET8 methyltransferase

Hitoshi Kurumizaka, Cheng-Han Ho, Yoshimasa Takizawa, Wataru Kobayashi, Yasuhiro Arimura, and Hiroshi Kimura

DOI: <https://doi.org/10.26508/lsa.202000919>

Corresponding author(s): Hitoshi Kurumizaka, The University of Tokyo

Review Timeline:

Submission Date:	2020-09-27
Editorial Decision:	2020-11-10
Revision Received:	2020-12-20
Editorial Decision:	2021-01-12
Revision Received:	2021-01-14
Accepted:	2021-01-14

Scientific Editor: Shachi Bhatt

Transaction Report:

November 10, 2020

Re: Life Science Alliance manuscript #LSA-2020-00919-T

Prof. Hitoshi Kurumizaka
The University of Tokyo
Institute for Quantitative Biosciences
1-1-1 Yayoi
Bunkyo-ku, Tokyo 113-0032
Japan

Dear Dr. Kurumizaka,

Thank you for submitting your manuscript entitled "Structural basis of nucleosomal histone H4 lysine 20 methylation by SET8 methyltransferase" to Life Science Alliance. The manuscript was assessed by expert reviewers, whose comments are appended to this letter.

As you will see from below, Reviewers 1 and 3 find your data intriguing and potentially important, but Reviewer 2 has expressed significant concerns about the claim that Set8 does not bind DNA on the NCP. However, we do think that there might be a path to published a revised version of this manuscript in Life Science Alliance (LSA) - that addresses all the minor concerns from all 3 reviewers and the request for better kinetics data and michaelis menten data on the various NCP preparations and the peptide (requested by R2). We understand that revisiting the preparation methods to better resolve the region of SET8 that is close to DNA might be out of scope for this manuscript, and can overrule that request, provided all other concerns raised during peer-review are addressed

Thank you for this interesting contribution to Life Science Alliance. We are looking forward to receiving your revised manuscript.

Sincerely,

Shachi Bhatt, Ph.D.
Executive Editor
Life Science Alliance
<https://www.lsjournal.org/>
Tweet @SciBhatt @LSAJournal

- A letter addressing the reviewers' comments point by point.
- An editable version of the final text (.DOC or .DOCX) is needed for copyediting (no PDFs).
- High-resolution figure, supplementary figure and video files uploaded as individual files: See our detailed guidelines for preparing your production-ready images, <https://www.life-science-alliance.org/authors>
- Summary blurb (enter in submission system): A short text summarizing in a single sentence the study (max. 200 characters including spaces). This text is used in conjunction with the titles of papers, hence should be informative and complementary to the title and running title. It should describe the context and significance of the findings for a general readership; it should be written in the present tense and refer to the work in the third person. Author names should not be mentioned.

B. MANUSCRIPT ORGANIZATION AND FORMATTING:

Reviewer #1 (Comments to the Authors (Required)):

Ho, Takizawa and colleagues report the cryo-EM structure of a SET8-nucleosome complex, explaining how SET8 selectively monomethylates histone H4 lysine-20 when H4 is nucleosome-

incorporated. In contrast to previous crystallographic studies, SET8 is not found to interact with DNA, but rather with the acidic patch in the nucleosome (either containing histone H3 or the centromeric CENPA variant). The arginine anchor in SET8 plays an important role in the monomethylation activity, as demonstrated by mutational analysis. The work is technically excellent, with structures solved to an average resolution of 3.15 and 3.00 Å, for the H3 and the CENPA nucleosome respectively. Although the local resolution of SET8 is lower, the acidic patch interactions are clearly resolved, resulting in a reliable and informative atomic model, which is validated biochemically.

In summary, this is a rigorous and informative study that deserves to be published, after addressing a number of minor points listed below.

1. Figure EV2 panels C and F show the cryoEM maps color coded according to their local resolution. A top view and a view tilted around the Y axis by 45 degrees are shown. These views make it difficult to clearly recognize SET8. I suggest a 90 degree tilt around the X axis would be more informative, similar to Figure 1D and 1E.

2. The authors explain the discrepancy between their cryoEM structure and the 4.5 Å resolution structure from Song Tan's group, in that SET8 was co-crystallised with a nucleosome containing the acidic patch interactor, RCC1. Could the authors elaborate more on whether they think DNA binding might be functionally relevant at some stage, perhaps on route to acidic patch engagement?

3. In the cryo-grid preparation methods, the authors could clarify whether any wait time was included before blotting and what blot force setting was used.

Reviewer #2 (Comments to the Authors (Required)):

Several inaccuracies are introduced in the text that should be revised

The authors mention "The H4K20 residue is mono-, di-, and trimethylated. H4K20 dimethylation and trimethylation are introduced by the methyltransferases Suv4-20h1/2, and are only promoted on monomethylated H4K20, but not on the unmethylated H4K20 residue". This statement is misleading as Dr. Wilson group showed that in vitro that Suv4-20h1/2 di-methylates K20. This section should be revised to reflect these findings (PMID : 24049080)

The author mentions that "Intriguingly, SET8 promotes H4K20 monomethylation only in the nucleosome, but not in nucleosome-free H4". Again this is inaccurate, several groups show that SET8 can methylate, albeit with less efficiency histone H4. This statement must be revised. (PMID : 15933070)

The following sentence does not read well at all. To test the functional importance of acidic patch binding by SET8, we prepared an acidic patch defective NCP (NCPapd), in which the acidic patch residues, H2A Glu56, H2A Glu61, H2A Glu64, H2A Asp90, H2A Glu91, H2A Glu92, H2B Glu105, and H2B Glu113, were replaced by Thr, Thr, Thr, Ser, Thr, Thr, Thr, and Thr, respectively (Kujirai et al, 2020). the authors must reformulate the last part sentence where an endless list of Thr is written.

The main reasons as to why I do not support publication is the lack of structural information for the region of the SET domain in close contacts with DNA which either suggest flexibility in that region or poor clustering of the particles. This major weakness leads the author to suggest that SET8 does not bind DNA on the NCP, which contradicts countless papers published in the last 15 years on the topic. The authors need to revisit the preparation methods to better resolve the region of SET8 that is close to DNA before making such claims.

Also, to make this manuscript worthy of publication in LS, the author need to provide better kinetics data and provide michealis menten data on the various NCP preparation (or octamer) as well as on the peptide. That would further the author's claim that SET8 does not require DNA to methylate the NCP.

Reviewer #3 (Comments to the Authors (Required)):

In this short manuscript, the authors reported the cryo-EM structures of SET8 bound to the H3 and CENP-A nucleosomes at ~3.0 Å resolution. They found that SET8, the enzyme that is responsible for methylation of H4K20, interacts with the acidic patch of the nucleosomes and do not have obvious interactions with DNA.

Overall, the study provides the structural basis for understanding how SET8 recognizes the nucleosome. The significant finding from this structural study is the role of acidic patch in Set8 recognition. However, this is not completely new as earlier mutation studies have already revealed its importance. Another interesting aspect of this study is that SET8 and DNA do not appear to have obvious interactions, which contradicts the earlier results based on a lower resolution (4.5 Å) crystal structure model. However, this point is not presented in a convincing way. By looking at the figures, it looks like SET8 is very close to DNA. If the authors want to claim no interactions between DNA and SET8, a figure that describes the interface between DNA and SET8 along with the local resolution of the density maps should be presented. Also, the authors should show a comparison between their structures and the earlier structural model. I understand the coordinate of the earlier model may not be available to the authors. If this is the case, the authors should indicate where the earlier proposed binding site is on the DNA in their structure. Finally, an uncertainty with the current cryo-EM study is that cross-linking is used to fix the complex, which needs to be discussed.

Reviewer #1 (Comments to the Authors (Required)):

Ho, Takizawa and colleagues report the cryo-EM structure of a SET8-nucleosome complex, explaining how SET8 selectively monomethylates histone H4 lysine-20 when H4 is nucleosome-incorporated. In contrast to previous crystallographic studies, SET8 is not found to interact with DNA, but rather with the acidic patch in the nucleosome (either containing histone H3 or the centromeric CENPA variant). The arginine anchor in SET8 plays an important role in the monomethylation activity, as demonstrated by mutational analysis. The work is technically excellent, with structures solved to an average resolution of 3.15 and 3.00 Å, for the H3 and the CENPA nucleosome respectively. Although the local resolution of SET8 is lower, the acidic patch interactions are clearly resolved, resulting in a reliable and informative atomic model, which is validated biochemically.

In summary, this is a rigorous and informative study that deserves to be published, after addressing a number of minor points listed below.

Reply)

Thank you very much for this favorable comment. We will revise the manuscript according to the points raised by this reviewer.

Comment 1)

Figure EV2 panels C and F show the cryoEM maps color coded according to their local resolution. A top view and a view tilted around the Y axis by 45 degrees are shown. These views make it difficult to clearly recognize SET8. I suggest a 90 degree tilt around the X axis would be more informative, similar to Figure 1D and 1E.

Reply)

In the revised manuscript, we presented a 90 degree tilt around the X axis, in addition to the previous figures.

Comment 2)

The authors explain the discrepancy between their cryoEM structure and the 4.5 Å resolution structure from Song Tan's group, in that SET8 was co-crystallised with a

nucleosome containing the acidic patch interactor, RCC1. Could the authors elaborate more on whether they think DNA binding might be functionally relevant at some stage, perhaps on route to acidic patch engagement?

Reply)

Thank you very much for this insightful comment. As suggested by this reviewer, the nucleosomal DNA binding of SET8 may be a stage on its route to the acidic patch engagement. In the revised manuscript, we extensively rewrote the second paragraph of the Discussion section, as below.

“The present structure differs substantially from the crystal structure model, in which SET8 binds nucleosomal DNA (Fig EV3). The previous crystallographic analysis was performed with the acidic patch binding protein, RCC1, to facilitate crystallization (Girish et al, 2016). The presence of RCC1 may inhibit proper SET8 arginine anchor binding to the nucleosomal acidic patch, and perturb active SET8-NCP complex formation. Meanwhile, the acidic patch blocking by RCC1 may help to capture a transient nucleosomal DNA binding state of SET8. This short-lived SET8-NCP binding may function in properly positioning SET8 on the NCP surface, where the SET8 arginine anchor and catalytic center bind to the acidic patch and the H4 N-terminal tail, respectively (Figs 1, 2, and 4).”

Comment 3)

In the cryo-grid preparation methods, the authors could clarify whether any wait time was included before blotting and what blot force setting was used.

Reply)

In the revised manuscript, we described the details of the cryo-grid preparation conditions, as suggested by this reviewer.

Reviewer #2 (Comments to the Authors (Required)):

Several inaccuracies are introduced in the text that should be revised

Reply)

Thank you very much for pointing these out. The inaccuracies mentioned by this reviewer have been adequately corrected in the revised manuscript, as described below.

Comment 1)

The authors mention "The H4K20 residue is mono-, di-, and trimethylated. H4K20 dimethylation and trimethylation are introduced by the methyltransferases Suv4-20h1/2, and are only promoted on monomethylated H4K20, but not on the unmethylated H4K20 residue". This statement is misleading as Dr. Wilson group showed that in vitro that Suv4-20h1/2 di-methylates K20. This section should be revised to reflect these findings (PMID : 24049080)

Reply)

Thank you very much for this comment. We missed this important work in the previous version of our manuscript. Now, we have accommodated this information with a new citation (Southall et al, 2014).

Comment 2)

The author mentions that "Intriguingly, SET8 promotes H4K20 monomethylation only in the nucleosome, but not in nucleosome-free H4". Again this is inaccurate, several groups show that SET8 can methylate, albeit with less efficiency histone H4. This statement must be revised. (PMID : 15933070)

Reply)

According to this suggestion, we revised the sentences in the Introduction and Discussion sections, as below.

Introduction:

"Intriguingly, SET8 primarily promotes H4K20 monomethylation in the nucleosome (Nishioka et al, 2002; Fang et al, 2002), although it also possesses the ability to

methylate the nucleosome-free H4K20 residue (Couture et al, 2005).”

Discussion:

“These findings explain why SET8 specifically monomethylates the nucleosomal H4K20 residue, rather than the nucleosome-free H4 (Nishioka et al, 2002; Fang et al, 2002).”

Comment 3)

The following sentence does not read well at all. To test the functional importance of acidic patch binding by SET8, we prepared an acidic patch defective NCP (NCP^{apd}), in which the acidic patch residues, H2A Glu56, H2A Glu61, H2A Glu64, H2A Asp90, H2A Glu91, H2A Glu92, H2B Glu105, and H2B Glu113, were replaced by Thr, Thr, Thr, Ser, Thr, Thr, Thr, and Thr, respectively (Kujirai et al, 2020). the authors must reformulate the last part sentence where an endless list of Thr is written.

Reply)

In the revised manuscript, we re-phrase the corresponding sentence as below.

“To test the functional importance of acidic patch binding by SET8, we prepared an acidic patch defective NCP (NCP^{apd}). In the NCP^{apd}, the acidic patch Glu and Asp residues (H2A Glu56, H2A Glu61, H2A Glu64, H2A Asp90, H2A Glu91, H2A Glu92, H2B Glu105, and H2B Glu113) were replaced by neutral, hydrophilic Thr and Ser residues, respectively (Kujirai et al, 2020).

Comment 4)

The main reasons as to why I do not support publication is the lack of structural information for the region of the SET domain in close contacts with DNA which either suggest flexibility in that region or poor clustering of the particles. This major weakness leads the author to suggest that SET8 does not bind DNA on the NCP, which contradicts countless papers published in the last 15 years on the topic. The authors need to revisit the preparation methods to better resolve the region of SET8 that is

close to DNA before making such claims.

Reply)

Thank you very much for this insightful comment. We agree that the DNA binding activity of SET8 may play an important role, probably by positioning the SET domain to interact with the proper nucleosomal site and bind to the acidic patch. In the revised manuscript, we extensively rewrote the corresponding second paragraph of the Discussion section, as below.

“The present structure differs substantially from the crystal structure model, in which SET8 binds nucleosomal DNA (Fig EV3). The previous crystallographic analysis was performed with the acidic patch binding protein, RCC1, to facilitate crystallization (Girish et al, 2016). The presence of RCC1 may inhibit the proper binding of the SET8 arginine anchor to the nucleosomal acidic patch, and perturb active SET8-NCP complex formation. Meanwhile, the acidic patch blocking by RCC1 may help to capture a transient nucleosomal DNA binding state of SET8. This short-lived SET8-NCP binding may function in properly positioning SET8 on the NCP surface, where the SET8 arginine anchor and catalytic center bind to the acidic patch and the H4 N-terminal tail, respectively (Figs 1, 2, and 4).”

Comment 5)

Also, to make this manuscript worthy of publication in LS, the author need to provide better kinetics data and provide michealis menten data on the various NCP preparation (or octamer) as well as on the peptide. That would further the author's claim that SET8 does not require DNA to methylate the NCP.

Reply)

As mentioned above, we now understand the importance of the DNA binding activity of SET8, and discussed it thoroughly. As the other reviewers also suggested, the DNA binding activity of SET8 may play a role to properly position the SET domain for the acidic patch engagement and the formation of the active SET8-nucleosome complex.

Therefore, in the revised manuscript, we do not stress the absence of the DNA binding, and focus more on the possible function of the DNA binding by SET8.

Reviewer #3 (Comments to the Authors (Required)):

In this short manuscript, the authors reported the cryo-EM structures of SET8 bound to the H3 and CENP-A nucleosomes at ~3.0 Å resolution. They found that SET8, the enzyme that is responsible for methylation of H4K20, interacts with the acidic patch of the nucleosomes and do not have obvious interactions with DNA.

Comment 1)

Overall, the study provides the structural basis for understanding how SET8 recognizes the nucleosome. The significant finding from this structural study is the role of acidic patch in Set8 recognition. However, this is not completely new as earlier mutation studies have already revealed its importance. Another interesting aspect of this study is that SET8 and DNA do not appear to have obvious interactions, which contradicts the earlier results based on a lower resolution (4.5 Å) crystal structure model. However, this point is not presented in a convincing way. By looking at the figures, it looks like SET8 is very close to DNA. If the authors want to claim no interactions between DNA and SET8, a figure that describes the interface between DNA and SET8 along with the local resolution of the density maps should be presented. Also, the authors should show a comparison between their structures and the earlier structural model. I understand the coordinate of the earlier model may not be available to the authors. If this is the case, the authors should indicate where the earlier proposed binding site is on the DNA in their structure.

Reply)

Thank you very much. This is an important and insightful comment. As this reviewer pointed out, the SET domain is located near the DNA in the SET8-nucleosome complex, although we could not observe the direct SET8-DNA interaction in the structures. This may be due to the low resolution around the SET8 region. We describe this fact at the end of the first paragraph of the Results section.

In addition, we revisited the earlier structural model, and found the possibility that these earlier structures may be transient states on the SET8 pathway to the nucleosomal acidic patch engagement, as suggested by the other reviewers. These new insights are discussed in the revised manuscript, as below.

“The present structure differs substantially from the crystal structure model, in which SET8 binds nucleosomal DNA (Fig EV3). The previous crystallographic analysis was performed with the acidic patch binding protein, RCC1, to facilitate crystallization (Girish et al, 2016). The presence of RCC1 may inhibit the proper binding of the SET8 arginine anchor to the nucleosomal acidic patch, and perturb active SET8-NCP complex formation. Meanwhile, the acidic patch blocking by RCC1 may help to capture a transient nucleosomal DNA binding state of SET8. This short-lived SET8-NCP binding may function in properly positioning SET8 on the NCP surface, where the SET8 arginine anchor and catalytic center bind to the acidic patch and the H4 N-terminal tail, respectively (Figs 1, 2, and 4).”

Comment 2)

Finally, an uncertainty with the current cryo-EM study is that cross-linking is used to fix the complex, which needs to be discussed.

Reply)

According to this suggestion, we described the uncertainty about the effects of cross-linking in the cryo-EM study, in the first paragraph of the Results section, as below.

“It is possible that the paraformaldehyde crosslinking may affect the SET8 interaction with the NCP.”

January 12, 2021

RE: Life Science Alliance Manuscript #LSA-2020-00919-TR

Prof. Hitoshi Kurumizaka
The University of Tokyo
Institute for Quantitative Biosciences
1-1-1 Yayoi
Bunkyo-ku, Tokyo 113-0032
Japan

Dear Dr. Kurumizaka,

Thank you for submitting your revised manuscript entitled "Structural basis of nucleosomal histone H4 lysine 20 methylation by SET8 methyltransferase". We would be happy to publish your paper in Life Science Alliance pending final revisions necessary to meet our formatting guidelines.

Along with the points listed below, please also attend to the following,

- please add a Running Title, Abstract, Summary Blurb/Alternate Abstract
- please use the [10 author names, et al.] format in your references (i.e. limit the author names to the first 10)
- please upload your Table file as an editable doc or excel file
- please add a callout for Figure S1 E-G and Figure S2 A,B,D,E,F to your main manuscript text

A. FINAL FILES:

-- Summary blurb (enter in submission system): A short text summarizing in a single sentence the study (max. 200 characters including spaces). This text is used in conjunction with the titles of

papers, hence should be informative and complementary to the title. It should describe the context and significance of the findings for a general readership; it should be written in the present tense and refer to the work in the third person. Author names should not be mentioned.

B. MANUSCRIPT ORGANIZATION AND FORMATTING:

Sincerely,

Shachi Bhatt, Ph.D.
Executive Editor
Life Science Alliance
<https://www.lsjournal.org/>
Tweet @SciBhatt @LSAJournal

Reviewer #1 (Comments to the Authors (Required)):

The authors have addressed my concerns in full and I think this manuscript should now be

accepted for publication.

Reviewer #3 (Comments to the Authors (Required)):

The revised manuscript has adequately addressed my concerns. I recommend its publication.

January 14, 2021

RE: Life Science Alliance Manuscript #LSA-2020-00919-TRR

Prof. Hitoshi Kurumizaka
The University of Tokyo
Institute for Quantitative Biosciences
1-1-1 Yayoi
Bunkyo-ku, Tokyo 113-0032
Japan

Dear Dr. Kurumizaka,

Thank you for submitting your Research Article entitled "Structural basis of nucleosomal histone H4 lysine 20 methylation by SET8 methyltransferase". It is a pleasure to let you know that your manuscript is now accepted for publication in Life Science Alliance. Congratulations on this interesting work.

DISTRIBUTION OF MATERIALS:

Again, congratulations on a very nice paper. I hope you found the review process to be constructive and are pleased with how the manuscript was handled editorially. We look forward to future exciting submissions from your lab.

Sincerely,

Shachi Bhatt, Ph.D.

Executive Editor

Life Science Alliance

<https://www.lsjournal.org/>
